

# Multi-frequency electrical impedance tomography as a non-invasive tool to characterize and monitor crop root systems

Maximilian Weigand[1] and Andreas Kemna[1]

[1]Department of Geophysics, University of Bonn, Meckenheimer Allee 176, 53115 Bonn, Germany

*Correspondence to:* Maximilian Weigand (mweigand@geo.uni-bonn.de)

**Abstract.** A better understanding of root-soil interactions and associated processes is essential in achieving progress in crop breeding and management, prompting the need for high-resolution and non-destructive characterization methods. To date such methods are still lacking, or restricted by technical constraints, in particular for characterizing and monitoring root growth and function in the field. A promising technique in this respect is electrical impedance tomography (EIT), which utilizes low-frequency ($< 1$ kHz) electrical conduction and polarization properties in an imaging framework. It is well established that cells and cell clusters exhibit an electrical polarization response in alternating electric current fields due to electrical double layers which form at cell membranes. This double layer is directly related to the electrical surface properties of the membrane, which in turn are influenced by nutrient dynamics (fluxes and concentrations on both sides of the membranes). Therefore it can be assumed that the electrical polarization properties of roots are inherently related to nutrient uptake and translocation processes in the roots. We here propose broadband (mHz to hundreds of Hz) multi-frequency EIT as a non-invasive methodological approach for the monitoring and physiological, i.e. functional, characterization of crop root systems. The approach combines the spatial resolution capability of an imaging method with the diagnostic potential of electrical impedance spectroscopy. The capability of multi-frequency EIT to characterize and monitor crop root systems was investigated in a laboratory rhizotron experiment, in which the root system of oilseed plants was monitored in a water-filled rhizotron under ongoing nutrient deprivation. We found a low-frequency polarization response of the root system, which enabled the successful delineation of the spatial extension of the root system. The magnitude of the overall polarization response decreased along with the physiological decay of the root system due to the nutrient deprivation. Spectral polarization parameters, as derived from a pixel-based Debye decomposition analysis of the multi-frequency imaging results, reveal systematic changes in the spatial and spectral electrical response of the root system. In particular, quantified mean relaxation times (of the order of 10 ms) indicate changes in the length scales on which the polarization processes took place in the root system, as a response to prolonged nutrient deficiency. Our results demonstrate that broadband EIT is a capable non-invasive method to image root system extension as well as to monitor changes associated with root physiological processes. Given its applicability at both laboratory and field scales, our results suggest an enormous potential of the method for the structural and functional imaging of root system for various applications. This particularly holds for the field scale, where corresponding methods are highly desired but to date lacking.





# 1 Introduction

Interest in and development of non-invasive methods for the structural and functional characterization and monitoring of root systems and the surrounding rhizosphere has substantially increased in recent years (e.g., Heřmanská et al., 2015, and references therein). This trend is driven mostly by the need to improve crop management and breeding techniques, and to reduce

fertiliser usage (e.g., Heege, 2013). In this context, various non-invasive methods for the investigation and characterization of crop root systems have been proposed (for a comprehensive overview of current methods, both for laboratory and field studies, see Mancuso, 2012). These methods include light transmission tomography (e.g., Pierret et al., 2003), X-ray computer tomography (e.g., Gregory et al., 2003; Pierret et al., 2003), neutron radiography (e.g., Willatt et al., 1978), magnetic resonance imaging (e.g., Metzner et al., 2015, and references therein), electrical resistivity tomography (ERT) (e.g., Mancuso,

2012), electrical capacitance measurements, and electrical impedance spectroscopy (EIS) (see Mancuso, 2012; Anderson and Hopmans, 2013, and references therein). However, most of these methods can not, or only under special circumstances, be used at the field scale, or they lack sensitivity to structural or physiological features of the rhizosphere (e.g., Mancuso, 2012).

Electrical methods, including both tomographic and spectroscopic approaches, are gaining importance among these methods due to their universal applicability at different scales and the recognized potential to provide pertinent information on root

systems via their electrical properties. Advances in measurement accuracy (Zimmermann et al., 2008) and large-scale deployments (e.g., Johnson et al., 2012; Loke et al., 2013) allow imaging studies with high spatial and temporal resolution at both laboratory and field scales (see, e.g., Kemna et al., 2012; Singha et al., 2014).

Electrical resistance measurements on root systems have been related to root age (Anderson and Higinbotham, 1976), to absorbing root surfaces of trees (Aubrecht et al., 2006; Čermák et al., 2006), and to surface area in contact with the ambient

solution (Cao et al., 2010). The measured resistances are usually interpreted by means of equivalent electrical circuit models of the root-soil continuum, and relations to biological properties are analyzed in terms of the circuit model parameters. Electrical imaging applications on crop root systems, however, are relatively rare. ERT has been used to map root zones (al Hagrey, 2007; Amato et al., 2008, 2009; al Hagrey and Petersen, 2011; Rossi et al., 2011) and to monitor water content in maize fields (Srayeddin and Doussan, 2009; Beff et al., 2013) and under an apple orchard (Boaga et al., 2013).

As pointed out by Urban et al. (2011), resistance methods for root characterization suffer from an inherent ambiguity of effective conductivity (or resistivity), making interpretation difficult. Polarization properties, on the other hand, provide valuable additional information, in particular if their spectral variation is explored. In geophysics, corresponding measurement approaches are referred to as induced polarization (IP) or spectral induced polarization (SIP) methods, since the polarization is provoked by an impressed electric field. A wide range of studies have investigated electrical polarization properties of plant

root systems, mostly in terms of capacitances, using alternating-current measurements at some frequency (e.g., Walker, 1965; Chloupek, 1972; Dvořák et al., 1981; Dalton, 1995; Aulen and Shipley, 2012; Dietrich et al., 2013). Correlations of varying strength have been found between measured capacitances and root (dry) mass, root surface, and various attributes associated with physiological processes such as root development. For example, Ellis et al. (2013) used an improved measurement setup to investigate the relation of electrical capacitances to root mass, root surface area, and root length in soil experiments. For



an overview of studies using electrical capacitance measurements on root systems, we refer to Kormanek et al. (2015). While in the above-mentioned studies single-frequency capacitance measurements were used, more recent studies also focused on the analysis of spectral measurements covering a broad frequency range, in terms of both capacitances (Ozier-Lafontaine and Bajazet, 2005) and impedances (Ozier-Lafontaine and Bajazet, 2005; Cao et al., 2011; Zanetti et al., 2011; Cseresnyés et al., 2013; Repo et al., 2014).

Research has also been conducted on electrical properties at the cellular scale, including electrical surface properties of cell walls and membranes (e.g., Kinraide, 1994; Wang et al., 2011). At an electrically charged surface in contact with an electrolyte an electrical double layer (EDL) forms (e.g., Lyklema, 2005). This EDL gives rise to electrical polarizability (e.g., Lyklema et al., 1983), that can be measured with EIS or electrical impedance tomography (EIT). Accordingly, variations in the EDL characteristics related to structural or functional changes in the root system should manifest in electrical impedance measurements.

Imaging of IP or SIP parameters has so far, to our knowledge, not been applied to the field of root research. However, various applications in near-surface petro- and biogeophysics have been successful. For example, spectral (i.e. multi-frequency) EIT was used to map subsurface hydrocarbon contamination at an industrial site (Flores Orozco et al., 2012a) and to monitor uranium precipitation induced by bacterial injections within the frame of contaminated site remediation (Flores Orozco et al., 2013)—both studies demonstrating the field-scale applicability of the method for subsurface (bio)geochemical characterization. Martin and Günther (2013) applied EIT to investigate fungus infestation of trees; however, in the imaging they did not take the spectral variation into account.

In the present work we propose broadband (mHz – kHz) multi-frequency EIT as an imaging tool for the physiological, i.e. functional, characterization of crop root systems. This novel approach for functional root imaging combines the spatial resolution benefits of EIT with the diagnostic capability of EIS, and builds upon instrumentation and processing tools that have been developed in recent years. Analogous to the now widely accepted interpretation of SIP signatures of soils and rocks in terms of textural and mineral surface characteristics, we hypothesize that the SIP response of crop root systems, which is imaged with the proposed methodology, is directly related to physico-chemical processes in the vicinity of electrical double layers forming in association with root physiological activity (e.g., nutrient uptake) at specific scales of the root system.

Besides the spatial delineation and monitoring of active root zones in terms of polarization magnitude, we aim at the analysis of the imaged SIP response in terms of relaxation times, which provides information on the spatial length scale at which the underlying processes occur. Relaxation times are determined using the Debye decomposition scheme, a phenomenological model that can describe a wide variety of SIP signatures (e.g., Nordsiek and Weller, 2008; Weigand and Kemna, 2016). A similiar procedure to analyse SIP signatures is also proposed by Ozier-Lafontaine and Bajazet (2005) for the analysis of SIP signatures measured on root systems.

To demonstrate the proposed methodology we conducted a laboratory experiment on oilseed plants grown in hydroponics. The plants were placed in a rhizotron container filled with tap water and monitored using multi-frequency EIT in the course of prolonged nutrient deficiency. The recovered spectral electrical signatures at various time steps were analyzed with regard



to total polarization strength and dominant relaxation time scales, and qualitatively related to the macroscopic reaction of the root system to the nutrient deprivation.

The next section shortly reviews electrical measurements on, and corresponding polarization properties of root systems. Then the geophysical methods used in the presented study are described, followed by the experimental setup and data acquisition/processing steps. The last two sections present the results and discuss methodological and biological aspects of the experiment.

## 2 Electrical properties and measurements of root systems

This section develops our working hypotheses regarding the electrical polarization of crop root systems. A more detailed description of the EDL is given and linked to the measurement methodology. We moreover shortly review previous works on small-scale (cells and cell suspensions) polarization of biomatter and the approaches used to analyze polarization measurements on whole root systems.

### 2.1 Electrical double layer polarization

Electrical conduction properties of soils are primarily determined by electrolytic soil water conductivity, i.e. ion concentration and mobility, and interface conduction processes at water-mineral interfaces. Electrical polarization properties originate mainly in ion accumulation processes in constrictions of the pore network and at water-mineral interfaces. If surfaces are electrically charged and in contact with an electrolyte, as for example found at mineral grain surfaces or cell membranes, electrical double layers (EDLs) form, which comprise the so-called Stern layer of bound counterions and the so-called diffusive layer. The latter is characterized by ion concentration gradients which result in equilibrium between electromigrative and diffusive ion fluxes. The EDL is affected by external electric fields, manifesting an induced polarization (IP), and takes a finite time (relaxation time) to reach equilibrium again once an impressed external field is turned off (e.g., Lyklema, 2005). Models of both Stern layer polarization (build-up of counterion concentration gradients in the Stern layer in the direction of the external electric field) (e.g., Schwarz, 1962; Leroy et al., 2008) and diffuse layer polarization (build-up of counterion and coion concentration gradients in the diffuse layer in the direction of the external electric field) (e.g., Dukhin et al., 1974; Fixman, 1980) have been developed, as well as models considering both Stern layer and diffuse layer (e.g., Lyklema et al., 1983; Razilov and Dukhin, 1995). In a porous system, such as soil, diffuse layer polarization is also referred to as membrane polarization since the resultant ion concentration gradients, for instance along a pore constriction, have an effect similar to an ion-selective membrane (e.g., Bücker and Hördt, 2013). Strength and relaxation behaviour of EDL polarization are, among other factors, influenced by background ion concentration in the pore water and surface charge density (e.g., Lyklema, 2005). Importantly, the relaxation time is related to the spatial length scale of the polarization process and the ionic diffusion coefficient in the EDL, which may be different for Stern layer and diffuse layer polarization (e.g., Lyklema et al., 1983). The relationship between relaxation time and characteristic length scale for induced polarization in soils and sediments has been investigated in many studies (e.g., Titov et al., 2002; Binley et al., 2005; Kruschwitz et al., 2010; Revil and Florsch, 2010; Revil et al., 2014).



## 2.2 Electrical measurements

5 Eletrical methods measure the conduction and polarization properties of a medium. In the frequency domain, the measured quantitiy is the complex-valued impedance, with the real (ohmic) part accounting for conduction, and the imaginary part accounting for polarization (capacitive) effects.

The electrical impedance, $\hat{Z}$, at some measurement (angular) frequency $\omega$ is defined as the ratio of the complex voltage $\hat{U}$ to the current $\hat{I}$, and can be represented by a real part $Z'$ and an imaginary part $Z''$:

$$10 \quad \hat{Z}(\omega) = \frac{\hat{U}(\omega)}{\hat{I}(\omega)} = Z'(\omega) + jZ''(\omega), \tag{1}$$

with $j$ denoting the imaginary unit. The inverse of the impedance is the admittance $\hat{Y}(\omega) = 1/\hat{Z}(\omega) = Y'(\omega) + jY''(\omega)$. Impedances, or admittances, can be translated to effective material properties by means of a (real-valued) geometrical factor $K$, which takes into account the geometric dimensions of the measurement (in particular electrode positions):

$$\hat{\rho}_a(\omega) = K\hat{Z}(\omega) = \frac{K}{\hat{Y}(\omega)}, \tag{2}$$

$$15 \quad \hat{\sigma}_a(\omega) = \frac{\hat{Y}(\omega)}{K} = \frac{1}{K\hat{Z}(\omega)} = \frac{1}{\hat{\rho}_a(\omega)}, \tag{3}$$

with $\hat{\rho}_a$ and $\hat{\sigma}_a$ being the apparent complex resistivity and apparent complex conductivity, respectively. These quantities are referred to as "apparent" because they do only represent the true properties if the medium under investigation is homogeneous. Otherwise they represent an effective (average) value. Spatial discrimination of electrical properties can be achieved by the use of multiple measurements with different electrode locations, which also form the basis for tomographic processing (inversion), 20 i.e. imaging.

Impedance measurements can be conducted using only two electrodes for a combined current and voltage measurement, or by using four electrodes (quadrupole measurements, also called four-point spreads) with separate current and voltage electrode pairs. In the latter case the contact impedance between electrode and medium, which becomes significant towards lower measurement frequencies, has practically no influence on the voltage measurement (e.g., Barsoukov and Macdonald, 2005).

## 25 2.3 Polarization of biomatter

Polarization phenomena of biomatter are commonly classified into three frequency regions with different polarization sources, namely the $\alpha$, $\beta$ and $\gamma$ regions (e.g., Schwan, 1957; Prodan et al., 2008). While overlapping, the low-frequency $\alpha$ polarization is thought to extend into the lower kHz range, followed by the $\beta$ polarization up to about 100 MHz, and joined by the $\gamma$ polarization at higher frequencies (e.g., Repo et al., 2012). Restricted by the mobility of the charge carriers, the $\alpha$-range is 30 assumed to be dominated by ionic polarization (i.e. the build-up and relaxation of ionic concentration gradients in an electric, time-variable field, as found in EDLs), the $\beta$-range by membrane polarization (e.g., Prodan et al., 2008), and the $\gamma$-range by molecular polarization. The different processes lead to different current flow paths within biomatter for different frequencies (Repo et al., 2012). These observations have been primarily made on cell suspensions and various kinds of tissue, which exhibit





structures much more homogeneous than fully developed plant and root systems. Polarization processes in plant roots are assumed to originate, among others, in the cell membranes, the apoplast and the symplast (Repo et al., 2014). The frequency-
dependence of published multi-frequency measurements (e.g., Ozier-Lafontaine and Bajazet, 2005; Repo et al., 2014) indicates multiple length scales, and associated structures, as the origin of electrical polarization responses.

On a cellular, or multi-cellular, level, much work has been conducted to gain information about the electrical surface characteristics. A Gouy-Chapman-Stern model relating surface charges to external ion concentrations has been formulated and subsequently improved (Kinraide et al., 1998; Kinraide, 1994; Wang et al., 2011). Using this model, ion activity at membrane
surfaces can be computed and analyzed for the investigation of physiological effects. These, and following, studies regarding ion toxicity and related surface electric potential have provided further evidence that certain surface potentials can be linked to physiological states and processes (for example, ion availability and uptake) (Wang et al., 2009; Kinraide and Wang, 2010; Wang et al., 2011, 2013). Recently, Li et al. (2015) determined Zeta-potentials of rice-root surfaces using elektrokinetic measurements, providing a relatively easy way to estimate surface potentials of intact plant roots.

The EDL is the source of polarization responses in the low-frequency range usually measured with EIS/EIT. It is sensitive to physiological processes that affect ion (nutrient) availability in the vicinity of, and ion fluxes across, charged cell walls and membranes. The key function of roots is the uptake of water and nutrients, which is highly dependent on nutrient availability, demand, and stress factors (e.g., Claassen and Barber, 1974; Delhon et al., 1995; Hose et al., 2001). Nutrient availability can influence water (and nutrient) transport within plant systems (e.g., Clarkson et al., 2000; Martínez-Ballesta et al., 2011),
and nutrient availability within roots can fluctuate in response to certain depletion situations (e.g., Benlloch-González et al., 2010). Also, the distribution of stress hormones such as ABA increases in response to stress situations, possibly inducing the aforementioned reactions (e.g., Schraut et al., 2005). The formation and properties of large-scale ion-selective structures such as endodermis and hypodermis are also directly influenced by the growth environment, and can change in response to external stress factors (Hose et al., 2001). In addition, Dalton (1995) noted that electrical polarization effects originate in the 'active'
parts of a root system only, which change according to age, nutrient availability, and other stress factors (see also Anderson and Hopmans, 2013).

The majority of studies concerning full root systems work with equivalent electrical circuit models to describe the measured signals of various biostructures (see Repo et al., 2012, and references therein). The scale and composition of these models vary considerably. For example, Dalton (1995) equates root segments with cylindrical capacitors, whose conducting plates are
formed by the inner xylem and the fluid surrounding the root segment, with the matter between acting as a dielectric. Kyle et al. (1999) proposed a simplified model of cell polarization in root systems, in which the cell membrane acts as a dielectric between the conducting inner and outer regions of the cells, thus representing a classical capacitor. Equivalent circuit representations inherently depend on the assumed flow paths of the electric current. For instance, impedance measurements using the stem as one pole for current injection and the medium surrounding the roots as the other pole (as frequently being done, e.g., Chloupek, 1976; Dietrich et al., 2013; Repo et al., 2014) force the current to cross all radial layers of the roots. However, even for stem injection, equivalent circuit models considerably simplify the true electrical processes in the root and root-rhizosphere system,
and it is questionable whether these models can be transferred between different experimental setups (as evident from the large





number of slightly different models that were proposed, e.g., Dalton, 1995; Ozier-Lafontaine and Bajazet, 2005; Dietrich et al., 2013). A purely phenomenological analysis is made by Repo et al. (2014), who use a classification approach to analyze spectral impedance data measured on pine roots infested with mycorrhizal fungi.

## 2.4 Working hypotheses

We propose to describe and interpret low-frequency (< 1 kHz) polarization processes in biomatter using concepts similar to those established for soils and rocks in recent years, under the assumption that the observed responses originate from the polarization of EDLs present in the biomatter. Accordingly, it should be possible to link the polarization magnitude to the average EDL thickness (which depends on the electric potential drop between the charged surface/membrane and the background ambient electrolyte), and characteristic relaxation times to the length scales at which the polarization processes

take place.

   Given the to-date observations and understanding of electrical polarization processes in biomatter, as reviewed in the previous section, our hypotheses are:

1. The magnitude of the low-frequency polarization response of roots is related to the overall surface area comprised by EDLs in the root-rhizosphere system, including the inner root structure. EDLs may form at Casparian strips (e.g.,
hypodermis and endodermis), cell walls and plasma membranes.

2. The characteristic relaxation times of the low-frequency polarization response of roots provide information on the length scales at which the polarization processes take place. While it is not clear to which extent a discrimination of specific polarization processes (e.g., cell wall polarization and polarization of the hypodermis) is possible, changes in the relaxation times should indicate changes in the length scale of the polarizing structures.

3. EDLs in the inner root system are influenced by ions (nutrients) in the sap fluid, EDLs at the outer root surface are influenced by ion concentrations in the external fluid. Thus, physiological processes that influence the availability, usage and translocation of ions directly influence the low-frequency polarization response.

4. Spectral EIT is a suitable non-invasive method to image and monitor magnitude and characteristic relaxation times of the low-frequency polarization response of root systems.

In the present study we address the second part of hypothesis 3, as well as hypothesis 4. Hypotheses 1 and 2 are based on the synthesis of existing works, but can neither be validated nor invalidated by the present study.

## 3 Material and methods

### 3.1 Electrical impedance tomography

The EIS (or SIP) method involves the measurement of impedances at multiple frequencies (usually in the mHz to kHz range).
It can be extended by utilizing electrode arrays consisting of tens to hundreds of electrodes to collect numerous, spatially



distributed four-point impedance data. From such data sets images of the complex conductivity (or its inverse, complex resistivity) can be computed using tomographic inversion algorithms (e.g., Kemna, 2000; Daily et al., 2005). This method is called complex conductivity (or complex resistivity) imaging, or electrical impedance tomography (EIT) and refers to both single- and multi-frequency (spectral) approaches. EIT images are characterized by a spatially variable resolution, which decreases

with increasing distance from the electrodes (e.g., Alumbaugh and Newman, 2000; Friedel, 2003; Binley and Kemna, 2005; Daily et al., 2005). The method has its primary fields of application in near-surface geophysics (e.g. Binley and Kemna, 2005; Daily et al., 2005; Revil et al., 2012) and medical imaging (e.g., Bayford, 2006).

Spectral EIT measurements presented in this study were conducted using the 40-channel EIT-40 impedance tomograph (Zimmermann et al., 2008), which was configured in a monitoring setup to acquire up to seven EIT data sets (frames) on a

mini-rhizotron container per day.

### 3.1.1   2D forward modeling

Synthetic impedance data, required in the tomographic inversion process, were modeled using the finite-element (FE) forward modeling code of Kemna (2000). The code solves the Poisson equation for a 2D complex conductivity distribution and 2D source currents in a tank of given thickness (Flores Orozco et al., 2012b). At the boundaries of the 2D modeling domain

now-flow Neumann conditions are imposed, which do not allow any current flow out of the modeling domain. Details of the implementation can be found in Kemna (2000).

A sketch of the FE grid (also used for the inversion and presentation of imaging results) resembling the rhizotron container is shown in Fig. 1b along with the position of 38 electrodes. The grid consists of 60 elements in $x$-direction, and 157 elements in $z$-direction (9,420 elements in total).

### 3.1.2   2D tomographic inversion

Complex conductivity images at multiple measurement frequencies were computed using the smoothness-constraint inversion code of Kemna (2000). The code computes the distribution of complex conductivity $\hat{\sigma}$ (expressed in either magnitude ($|\sigma|$) and phase ($\phi$), or real component ($\sigma'$) and imaginary component ($\sigma''$)), in the 2D ($x,z$) image plane from the given set of complex transfer impedances $\hat{Z}_i$ (expressed in magnitude ($|\hat{Z}_i|$) and phase ($\varphi_i$)) under the constraint of maximum model smoothness. Log-transformed impedances and log-transformed complex conductivities (of the individual elements of the grid) are used as data and model parameters, respectively, in the inversion. The iterative, Gauss-Newton-type inversion scheme minimizes an objective function composed of measures of data misfit and model roughness. The data misfit is weighted by individual data errors, which are computed using the resistance error model (LaBrecque et al., 1996)

$$\Delta|\hat{Z}_i| = a|\hat{Z}_i| + b, \tag{4}$$

with $\Delta|\hat{Z}_i|$ being the error of impedance magnitude (resistance) $|\hat{Z}_i|$, $a$ the relative error contribution and $b$ the absolute error contribution of impedance magnitude. For more details on the inversion scheme we refer to Kemna (2000). The inversion is performed for each frequency of the given data set separately.



## 3.2 Debye decomposition

The Debye decomposition (DD) approach (e.g., Uhlmann and Hakim, 1971; Lesmes and Frye, 2001) was used to analyze the complex conductivity spectra recovered from the multi-frequency EIT inversion results. The approach yields integral parameters describing the spectral characteristics of the SIP signature. The complex conductivity spectrum is represented as a superposition of a large number of Debye relaxation terms at relaxation times $\tau_k$ (suitably distributed over the range implicitly defined by the data frequency limits, see Weigand and Kemna (2016)):

$$\hat{\sigma}(\omega) = \sigma_\infty \left[ 1 - \sum_k \frac{m_k}{1 + j\omega\tau_k} \right], \tag{5}$$

with $\sigma_\infty$ being the (real-valued) conductivity in the high-frequency limit, and $m_k$ the $k$-th chargeability, describing the relative weight of the $k$-th Debye relaxation term in the decomposition. The chargeabilities $m_k$ at the different relaxation times $\tau_k$ form a relaxation time distribution (RTD), from which the following descriptive parameters are computed (e.g., Nordsiek and Weller, 2008):

- The normalized total chargeability $m_{\mathrm{tot}}^{\mathrm{n}}$ is a measure of the overall polarization reflected in the spectrum (e.g., Tarasov and Titov, 2013; Weigand and Kemna, 2016):

$$m_{\mathrm{tot}}^{\mathrm{n}} = \sigma_0 \sum_k m_k, \tag{6}$$

  with $\sigma_0$ being the (real-valued) conductivity in the low-frequency limit.

- The mean logarithmic relaxation time $\tau_{\mathrm{mean}}$ represents a weighted mean of the RTD:

$$\tau_{\mathrm{mean}} = \exp\left( \frac{\sum_k m_k \log(\tau_k)}{\sum_k m_k} \right). \tag{7}$$

- The uniformity parameter $U_{60,10}$ describes the frequency dispersion of the spectrum:

$$U_{60,10} = \frac{\tau_{60}}{\tau_{10}}, \tag{8}$$

  with $\tau_{10}$ and $\tau_{60}$ being the relaxation times at which the cumulative chargeability reaches 10 % and 60 %, respectively, of the total chargeability sum.

The implementation of Weigand and Kemna (2016) was used for the DD analysis. The iterative inversion scheme balances between (error-weighted) data fitting and smoothing requirements.

## 3.3 Experimental setup

### 3.3.1 Rhizotron

The experiment was conducted using a mini-rhizotron container with the dimensions of 30 cm width, 78 cm height, and 2 cm depth, and a transparent front plate (Fig. 1). The front of the rhizotron is equipped with 38 brass pins of 5 mm diameter as





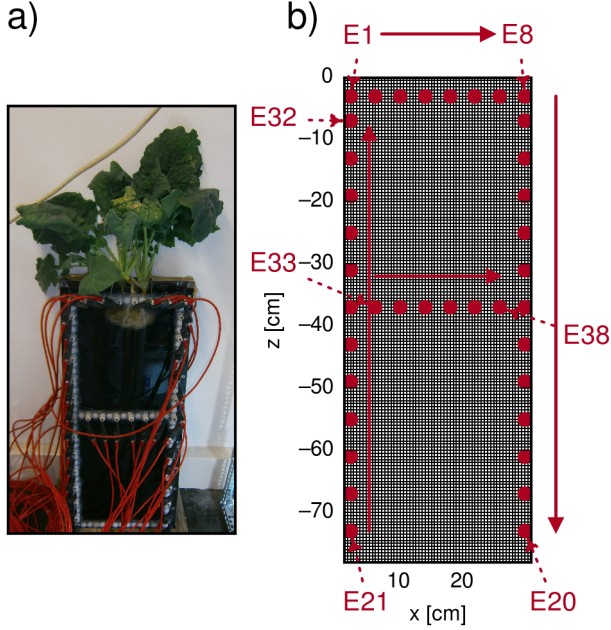

**Figure 1.** a) Experimental setup of plant root systems in water filled rhizotron. b) Corresponding finite-element grid used for electrical modeling and imaging. Red dots indicate position of electrodes.

electrodes, which do not extend into the rhizotron inner volume. A growth lamp was installed above the rhizotron and turned on during daylight hours.

### 3.3.2 Plant treatment

Oilseed plants had been grown in nutrient solution prior to the experiment. To increase the root mass, three plants were tied together and centrally placed at the top of the rhizotron (Fig. 1), which before had been filled with tap water. No water was added to the rhizotron during the experiment.

### 3.3.3 Data acquisition

Over the course of 3 days 21 EIT data sets at 35 frequencies between 0.46 Hz and 45 kHz were collected in regular intervals, starting right after the placement of the plant system in the rhizotron. A total of 1,158 quadrupoles were measured for each data set, involving 74 individual current injections (plus 767 reciprocal configurations, where current and voltage electrode pairs are interchanged, for quality assessment), requiring less than four hours acquisition time. These quadrupoles consisted mostly of skip-0 and skip-2 (numbers of electrodes between the two electrodes used for current injection and voltage measurement, respectively) dipole-dipole configurations, as well as quadrupoles with current electrodes on opposite sides (left and right) of the rhizotron and skip-0 voltage readings.



### 3.4 Data processing

#### 3.4.1 Selection of impedance data

The inversion scheme assumes normally distributed and uncorrelated data errors and is very sensitive to outliers (e.g., LaBrecque and Ward, 1990). Outliers are usually associated with low signal-to-noise ratios or systematic errors due to missing or bad electrode contacts. Outliers can either be removed from the data set prior to inversion, or accounted for by sophisticated, 'robust' inversion schemes (LaBrecque and Ward, 1990). In these robust schemes, the weighting of individual data points is iteratively adapted, which can lead to a reduction of spatial resolution as well as recovered contrast in the imaging results. However, usually this does not change the qualitative results of the inversion. In the present study we sought to analyze data across the frequency and time domains, which requires a careful and consistent analysis of the inversion data. Thus, to prevent introducing unnecessary variations between time-steps and frequencies, we opted to remove outliers using the criteria described below and use individual, but consistent, data weighting schemes for all measurements.

The measured impedance data (also referred to as 'raw data', in contrast to complex conductivity data recovered from the imaging results, referred to as 'intrinsic data') were screened (filtered) for outliers and faulty data according to multiple criteria: First, outliers were identified for each frequency and time step and removed from the data set. Due to the underlying physical principles, EIT measurements usually do not show strong variations when electrode positions are only slightly shifted. The exception here are measurements with electrodes located close to the plant stem system, where a very localized anomalous response was expected in the data. Accordingly, care was taken not to remove these data as outliers. Following this selection process, only impedance spectra were kept that retained more than 90 % of the original data points below 300 Hz and showed consistency over several time steps. To avoid errors due to electromagnetic coupling effects (e.g., Pelton et al., 1978; Zhao et al., 2013), only data below 220 Hz were considered for the imaging. Measurements at 50 Hz were discarded due to powerline noise.

The applied data selection criteria resulted in small variations of the number of measurements actually used for the inversions for the different time steps, ranging between 530 and 555 measurements per data set and frequency. The average injected current strength of the measurements at each time step increased slightly over time from approximately 1.0 mA to 1.2 mA.

#### 3.4.2 Correction of impedance data for imperfect 2D situation

Since the electrodes do not extend across the entire depth (i.e. horizontal direction perpendicular to the image plane) of the rhizotron the electric current and potential field distributions in the rhizotron are not perfectly 2D, as is assumed in the forward modeling. Therefore measurements were conducted on a rhizotron solely filled with tap water of known conductivity. By comparing the latter with the apparent conductivity (eq. (3)) derived from the measured impedance and the numerically determined geometric factor (obtained from running the forward model for a homogeneous case) for each measurement configuration, correction factors were computed and applied to all measured impedances.

In Fig. 2 the effect of this correction procedure on the EIT inversion result is shown. Without correction the obtained image exhibits an artificial pattern (Fig. 2a), while with correction a practically homogeneous distribution is recovered, in agreement with the conductivity of the tap water (Fig. 2b).





The inversion was conducted using the error parameter values $a = 0.5$ % and $b = 0.012$ $\Omega$ (cf. eq. (4)). These values were found to be appropriate and were also used in the inversions of which the results are shown in the following.

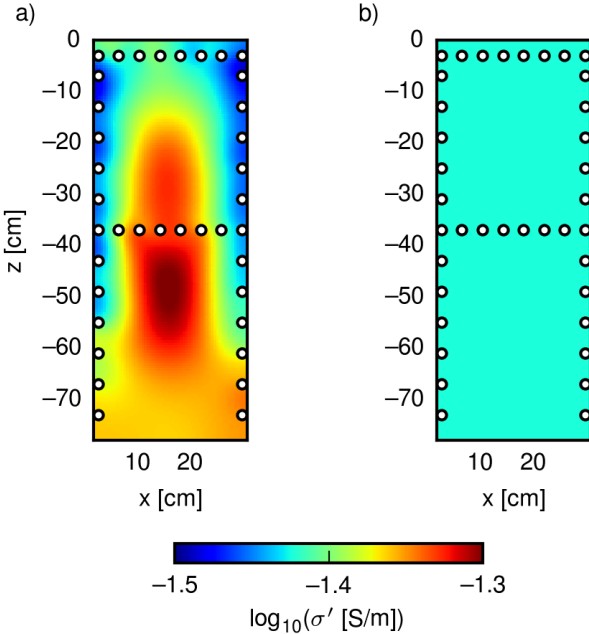

**Figure 2.** EIT inversion results (real component of complex conductivity) for measurements on a rhizotron filled with water of known conductivity (375 µS/cm = $10^{-1.43}$ S/cm) without (a) and with (b) correction of the impedance data for the imperfect 2D situation.

### 3.4.3    Adaptation of modeling domain to changing water table

Due to evaporation and root water uptake the water table fell by ca. 2 cm over the course of the monitoring experiment. This was not problematic in terms of electrode contact as electrodes always remained in the water. However, the changing water

table has to be accounted for in the EIT inversion by means of an adapted modeling domain, i.e. by adapting the position of of the top boundary of the FE modeling grid, where no-flow conditions are assumed. Otherwise, as we checked in numerical experiments, significant artefacts appear in the inversion results (Fig. 3).

From the known water tables at the beginning and end of the experiment, and the average time when each EIT data set was collected, the positions of the top boundary of the individual grids used for the inversion of each data set were determined by

linear interpolation.

### 3.4.4    Analysis of spectral imaging results

The spectral imaging results were analyzed by means of pixel-wise application of the Debye decomposition scheme. As water exhibits no significant polarization response in the examined frequency range, the area free of roots from 20 cm to 78 cm depth




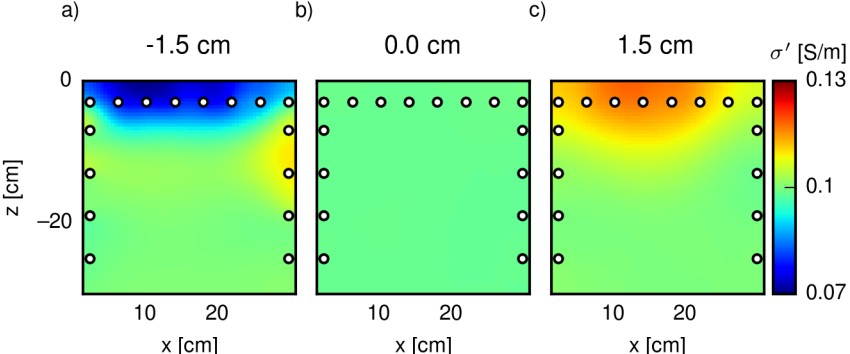

**Figure 3.** EIT inversion results (real component of complex conductivity) obtained from synthetic data using a modeling grid in the inversion with 1.5 cm lower (a), identical (b), and 1.5 cm higher (c) position of the top boundary compared to the grid used to simulate (forward model) the data. The forward model was homogeneously parameterized with a conductivity distribution of 0.1 S/m. Only the upper part of the modeling domain (rhizotron) is shown. Electrode positions and measurement configurations are the same for all three cases.

of the rhizotron was used to quantify a $m_{\mathrm{tot}}^{\mathrm{n}}$ threshold value below which polarization is considered insignificant. Based on this threshold value the entire images of spectral parameters obtained from the Debye decomposition, i.e. including the top 20 cm of the rhizotron, were partitioned into pixels with and without significant polarization. The observed polarization can be fully attributed to the root system (no polarization is expected from the surrounding water in the examined frequency range) and thus the corresponding pixels delineate polarizable areas of the root system, which we refer to the root pixel group.

To analyze the temporal evolution of the overall root system polarization (in terms of normalized total chargeability) the root pixel group was determined for the first time step, and then kept fix for the following time steps. Relaxation times, however, can only be reliably extracted from SIP signatures if they show significant polarization. Therefore, for the relaxation time analysis (in terms of mean relaxation time and uniformity parameter) the root pixel groups were determined for each time step individually.

# 4 Results

## 4.1 Physiological response

Photographs of the plant systems at the beginning and the end of the experiment are shown in Figure 4. As is evident from the photographs, the plants significantly reacted to the nutrient stress situation and degraded over time. The root systems extended down to a depth of approximately 13 cm (cf. Fig. 1).



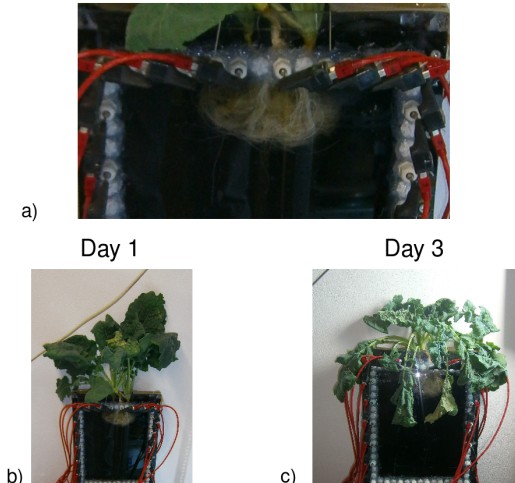

**Figure 4.** Photographs of the oilseed plants during the experiment: a) close-up of the root systems in the rhizotron container, b) day 1, c) day 3.

## 4.2    Impedance spectra

In Figure 5 the temporal evolution of the raw data spectra in terms of apparent complex conductivity (eq. (3)) is shown for two exemplary measurement configurations: a quadrupole with electrode pairs on both sides of the rhizotron located directly above the root system, i.e. with sensitivity to the root system, and a quadrupole with electrodes from the horizontal electrode line at 37 cm depth, i.e. located relatively far away from the root system and thus sensitive only to the water. The real component of apparent complex conductivity shows a smooth, consistent behaviour across the time and frequency domains for both "with roots" and "water-only" responses (Figs. 5a,b). However, the conductivity decreases for the quadrupole around the root system, while it increases in the 'water-only' quadrupole. The imaginary components, i.e. the polarization responses, with roots (Fig. 5c) are also consistent, and show changes, especially in the lower-frequency range, over time. The water-only measurements, on the other hand, exhibit only negligible polarization responses, likely dominated by measurement errors and noise (Fig. 5d). The polarization magnitudes, on one hand, lie well below the signal threshold that can be reliably measured with the EIT-40 system (Zimmermann et al., 2008). The jittery shape of these signatures (Fig. 5d) is attributed to the logarithmic scale of the plot. On the other hand, measured root signatures lie clearly above the measurement threshold of the system (Fig. 5c).



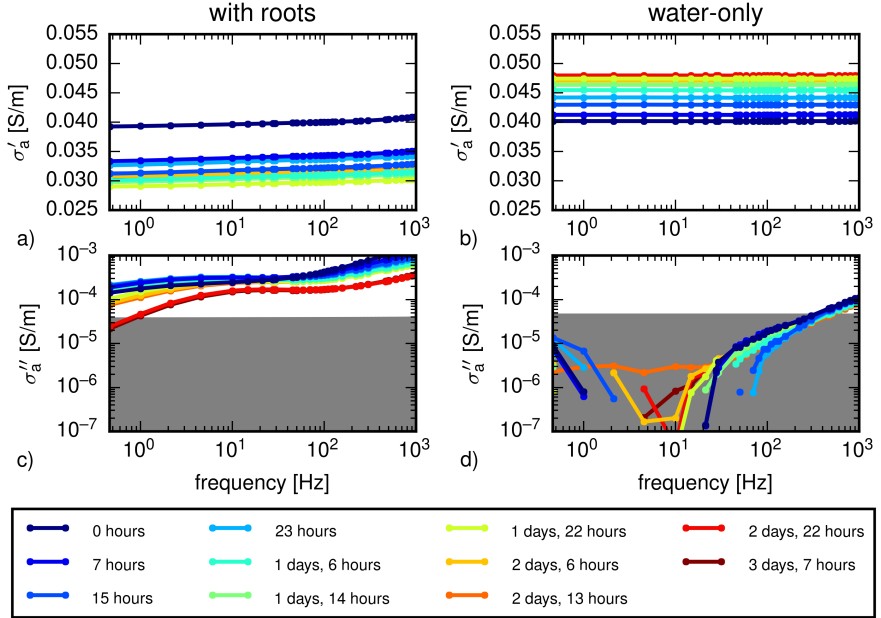

**Figure 5.** Temporal evolution of raw data spectra, plotted as real (a,b) and imaginary (c,d) components of apparent complex conductivity (eq. (3)), for two exemplary measurement configurations: a,c) current injection between electrodes 3 and 4 and voltage measurement between electrodes 5 and 6 (quadrupole located directly above the root system, i.e. response "with roots"); b,d) current injection between electrodes 34 and 35 and voltage measurement between electrodes 36 and 37 (quadrupole located in an area relatively far away from the root system, i.e. "water-only" response). For electrode numbering, see Fig. 1b. Blue color indicates early measurements, while later ones are shown in red. Values of $\sigma''$ that lie below the measurement accuracy of the system (1 mrad phase shift at 1 kHz for water, see Zimmermann et al. (2008)) are indicated by gray areas.

## 4.3 Single-frequency imaging results

The spatial variability of the electrical response was assessed using the complex conductivity imaging results, i.e. $\sigma'$ and $\sigma''$, at the first time step for the two frequencies 1 Hz and 70 Hz (Fig. 6). Only weak variations in the real component (in-phase conductivity) can be observed at the location of the root system (Fig. 6b,d). However, a significant polarization response in the imaginary component (Fig. 6c,e) coincides with the extension of the root system. The frequency dependence previously found in the apparent complex conductivity spectra (cf. Fig. 5) is also revealed in the imaging results, with a stronger response at 70 Hz than at 1 Hz. It manifests both in signal strength and in the spatial extension of the polarizable anomaly associated with the root system.





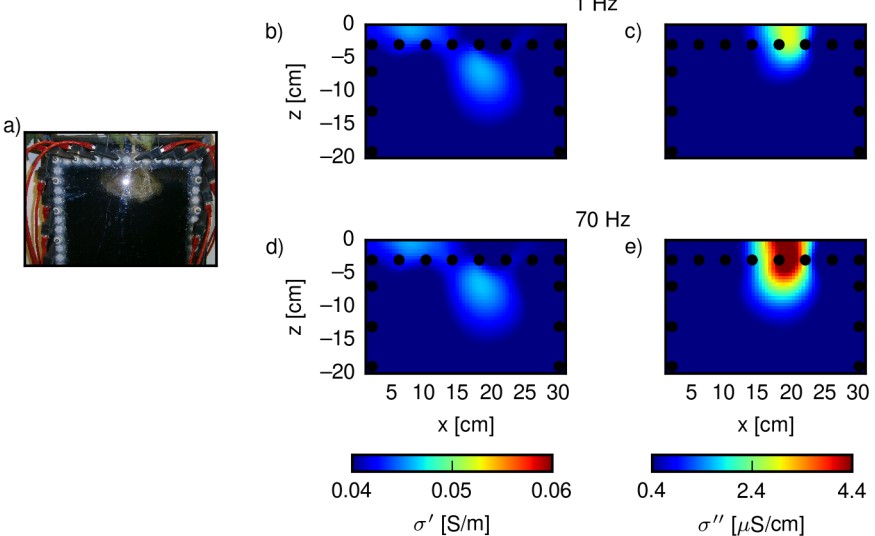

**Figure 6.** Single-frequency inversion results in terms of real (b,d) and imaginary (c,e) components of complex conductivity for 1 Hz (b,c) and 70 Hz (c,e) at the first time step. The photograph of the root system at this time (a) shows the same area of the rhizotron as the inversion results.

## 4.4 Complex conductivity spectra recovered from imaging results

Complex conductivity spectra were extracted from the multi-frequency imaging results at three locations: near the stem area of the root system, from the lower area of the root system, and from the lower half of the rhizotron, where no roots were present. These locations thus represent areas with, respectively, relatively large roots, small roots, and no roots at all. Figure 7 shows the temporal evolution of the spectral response for the three locations. The real component of complex conductivity ($\sigma'$) increased over the course of the experiment for all three locations (Fig. 7a-c). The imaginary component of complex conductivity ($\sigma''$) reveals a frequency-dependent polarization response at the root locations for all time steps (Fig. 7d,e). The polarization magnitude decreases over time, and changes in the shape of the spectra can be observed for later time steps. These changes are most pronounced for the location near the stem (Fig. 7a). The polarization signatures recovered at the bottom of the rhizotron (water only) show almost two orders of magnitude smaller magnitudes (Fig. 7f), compared to those in the root areas; they are more noisy and do not exhibit a clear frequency trend.

## 4.5 Debye decomposition of recovered complex conductivity spectra

The Debye decomposition scheme was applied to the complex conductivity spectra recovered from multi-frequency EIT to quantify the overall polarization (normalized total chargeability $m_{tot}^{n}$) and the characteristic relaxation time (mean relaxation time $\tau_{mean}$), as well as the uniformity parameter $U_{60,10}$. By means of this analysis, the intrinsic spectra can be assessed with respect to the magnitude and shape of the polarization response for all pixels at each time step. Figure 8 shows a decomposition





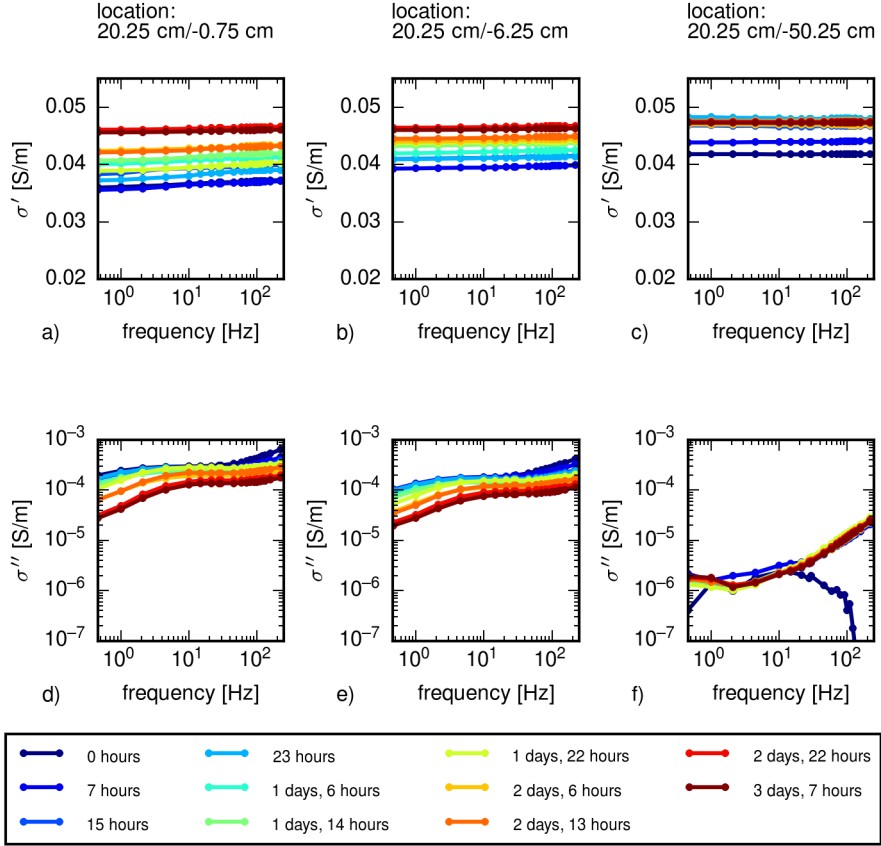

**Figure 7.** Intrinsic complex conductivity spectra (in terms of real component $\sigma'$ and imaginary component $\sigma''$) for all time steps (indicated by colour of the curves) recovered from the multi-frequency inversion results at different locations in the rhizotron: a,d) stem area; b,e) bottom of the root system; c,f) water-only location.

result for a pixel from the stem area for the first time step, corresponding to the spectrum plotted in Fig. 7a,d at "0 hours" (dark blue curve). The complex conductivity spectrum was fitted by means of 96 Debye relaxation terms (Fig. 8a), yielding a relaxation time distribution (RTD) (Fig. 8b), from which $\tau_{\mathrm{mean}}$, $\tau_{10}$ and $\tau_{60}$ can be determined (Fig. 8a,b). We note that $\tau_{\mathrm{mean}}$ does not coincide with the RTD peak, which only happens if the RTD shows a perfect symmetry (in log scale), which is not the case here.



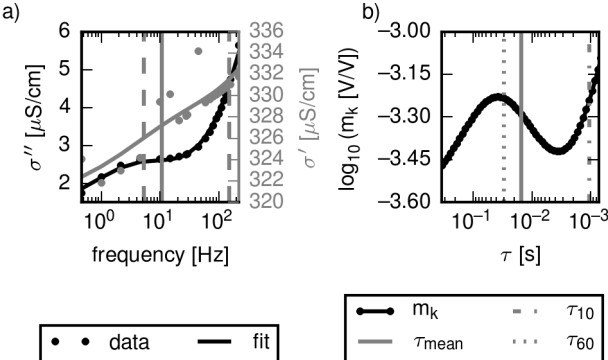

**Figure 8.** Debye decomposition of the recovered complex conductivity spectrum for a pixel from the stem area with maximum polarization response (cf. Figs. 5,7): a) complex conductivity (gray: real component, black: imaginary component) from spectral EIT (dots) and fitted DD response (solid curves); b) corresponding relaxation time distribution. Vertical gray solid lines indicate $\tau_{\mathrm{mean}}$, and the dashed vertical lines indicate $\tau_{10}$ and $\tau_{60}$, respectively.

## 4.6   Images of spectral parameters obtained from Debye decomposition

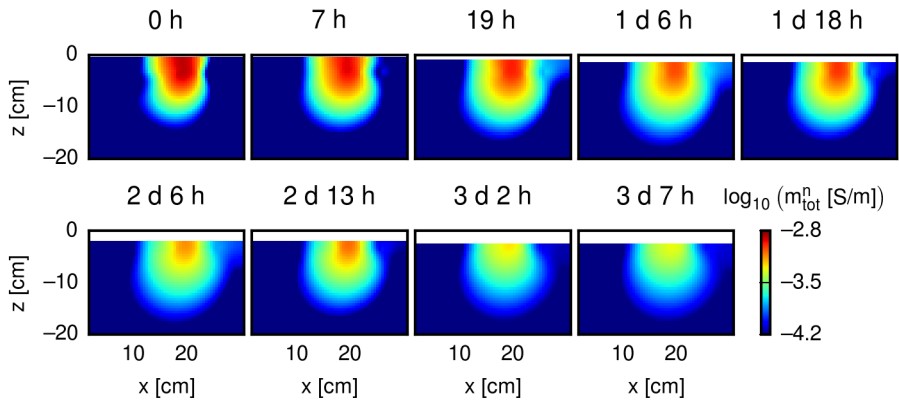

**Figure 9.** Spatial distribution of DD-derived parameter $m_{\mathrm{tot}}^{\mathrm{n}}$ for selected time steps. The top boundary is adjusted according to the estimated water table for each measurement time.

Images depicting the DD-derived total polarization ($m_{\mathrm{tot}}^{\mathrm{n}}$) results of the complex conductivity spectra (obtained from multi-frequency EIT) for selected time steps are presented in Figure 9. The extension of the root system against the surrounding water (characterized by low polarization) is clearly delineated in the images, and a continuous decrease in polarization strength is observed over time.

For further analysis, the complex conductivity spectra (also referred to as pixel spectra) were classified into two categories, with and without roots, as described in Section 3.4.4. The resulting "root spectra" were then processed separately, and care was





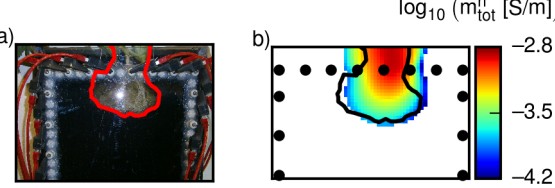

**Figure 10.** Comparison of root extension inferred from the photograph (a), indicated by overlaid solid lines, and $m_{\mathrm{tot}}^{\mathrm{n}}$ results (b) for the first time step. Plotted in b) are only pixels from the root zone i.e. pixels with a polarization response above the identified $\sigma''$ threshold value.

taken that the selected spectra exhibit a sufficiently strong and consistent polarization response to allow a reliable relaxation time analysis. Figure 10 shows the comparison of the $m_{\mathrm{tot}}^{\mathrm{n}}$ results for the first time step with the extension of the root system according to the photograph. The root area reconstructed from the spectral EIT results shows a good agreement with the known outer boundaries of the root system. Systematic changes in the overall root system response were analyzed by averaging the

$m_{\mathrm{tot}}^{\mathrm{n}}$ pixel values in the root zone (Fig. 11). This average polarization response shows a steady decrease over time.

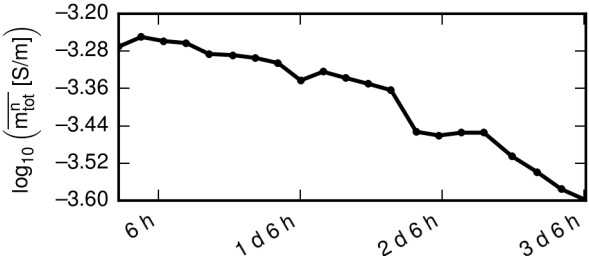

**Figure 11.** Mean value of DD-derived total chargeability $m_{\mathrm{tot}}^{\mathrm{n}}$ plotted versus time after start of the experiment. Average values were computed based on all pixels belonging to the root zone.

Images of the DD-derived mean relaxation time $\tau_{\mathrm{mean}}$ are presented in Fig. 12 for selected time steps. Spatial variations within the root zone can be observed for each time step, as well as changes between time steps. Noticeable is a general trend from larger relaxation times (up to 18 ms) to smaller relaxation times (down to 9 ms) over the course of the experiment. Corresponding images of the uniformity parameter $U_{60,10}$ are shown in Fig. 13. Observed variations within images and between

time steps indicate changes in the shape of the pixel spectra. Values approaching 1 indicate a stronger spectral dispersion, i.e. a focusing of the spectral polarization response in a narrower frequency band.



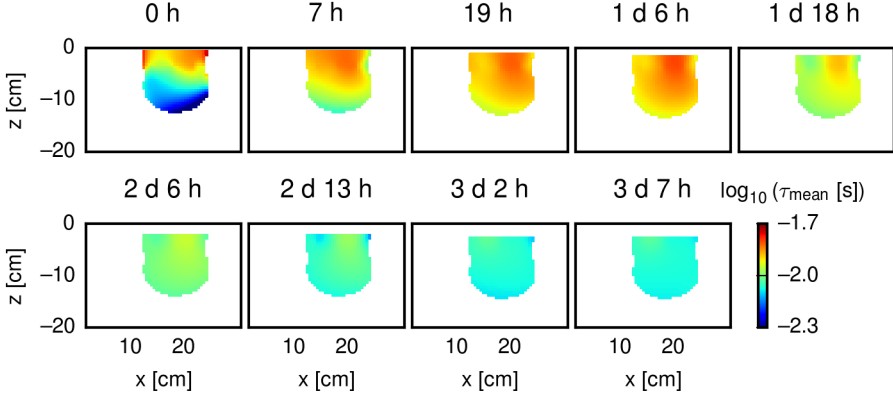

**Figure 12.** Spatial distribution of the DD-derived parameter $\tau_{\mathrm{mean}}$ for selected time steps. Only pixels belonging to the root zone are plotted. Masked (white) pixels were classified as water. The top boundary is adjusted according to the estimated water table for each measurement time.

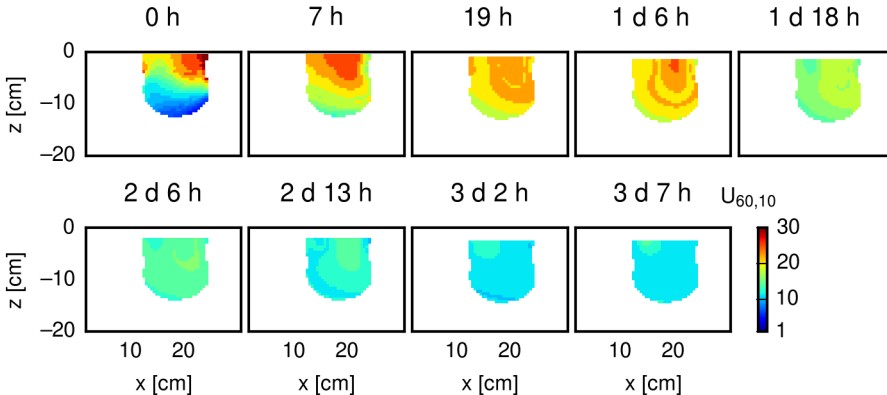

**Figure 13.** Spatial distribution of the DD-derived parameter $U_{60,10}$ for selected time steps. Only pixels belonging to the root zone are plotted. Masked (white) pixels were classified as water. The top boundary is adjusted according to the estimated water table for each measurement time.

## 5   Discussion

The following discussion is divided into two parts: the biological discussion of the experiment, and the assessment of the geophysical methodology for crop root investigations.

### 5.1   Biological interpretation

The conductivity in the rhizotron increased with time (Fig. 7a-c), which we explain with increased salinity due to evaporation and root water uptake (see also supplement 1, which provides images of the conductivity for selected time steps). It is possible



that residual nutrient solution with a high salinity was brought into the rhizotron adhering to the roots, although they were washed before they were placed in the rhizotron. The increase in conductivity observed in the raw data for the measurement

surrounding the root system (Fig 5a) is probably caused by the increased (more resistive) root volume in the sensitive area of the measurement due to the dropping water table.

The physiological response of the root system to the imposed nutrient deprivation is reflected by a decreasing overall polarization response (Figs. 9 and 11). Note that it is highly unlikely that the dropping water table caused this decrease in polarization, as more current is forced through the root system with the dropping water table, which should actually increase

the polarization response, given that this response does not change due to physiological reactions in the root system . We attribute this decrease in polarization to a general weakening of the EDLs present in the root system. The cause of this EDL weakening can be manifold and can not be isolated in this study. Plant-root systems represent a hydraulically connected system whose water potential is primarily controlled by water evaporation at the leaves. In case of intact hydraulic connectivity in the plant, a decrease in water potential due to evaporation causes water and solute uptake by the roots and water and nutrient

flow from the roots to the leaves (Tinker and Nye, 2000). Accordingly, it can be expected for our experiment that the outer root regions were depleted of nutrients first, as the available nutrients were translocated to the stem and the leaf areas. As a result, the ion concentration decreased and the EDLs in these depleted areas were weakened, implying a decreased polarization response. Following the reasoning, the stem should, however, retain more nutrients over a longer time span, resulting in a more robust polarization response under nutrient deprivation in this area than further away from the stem. This is consistent with the

observed time-lapse imaging results (Fig. 9). Plant reactions to nutrient deprivation, however, can manifest in more ways than the pure translocation of available nutrients, and other processes such as reduced metabolic activity or nutrient accumulation in roots may play an important role in the characteristics of the root electrical properties, too. Especially, if dynamic responses are considered, plants can sense, and react to, stress situations and can initiate changes in solute transport and hydraulic membrane conductivity properties (e.g., Clarkson et al., 2000; Schraut et al., 2005).

Another indication of the physiological stress response are the changes in the shape of the spectra (Figs. 12 and 13). Relative changes in the relaxation time contributions suggest changes in the underlying structures that control the polarization response at certain time steps. These changes might be related to new or ceased ion fluxes and their varying pathways within the root systems, as well as to varying surface charges at various structures such as the endodermis. If these structures change, or break down, in response to nutrient deprivation, corresponding changes in the electrical properties can be expected. However, given

its spatial resolution limits, EIT does not allow to distinguish these different structures.

Assuming that relaxation times can be linked to length scales of the underlying polarization processes, the observed signatures indicate multiple polarizable structures. However, the methodology applied here prevents further investigations in this direction. In contrast to most of the existing studies, we did not inject current directly in the stem, and correspondingly the explicit current pathways are much less defined in our approach. This prevents (at this stage) a simple formulation of an equiv-

alent lumped electrical circuit model. Comparison of measurements using the procedure presented here with a stem-injection approach could, however, help elucidating the origin of polarization and its length-scale characteristics. Current injection into the stem forces the current to flow through the root system and through all radial layers of the roots, and thus a stronger polar-





ization response from inside the root can be expected, as well as the polarization of additional membrane structures. Additional experiments could focus on establishing relationships between recovered spectral polarization parameters and root specific

parameters, such as surface area and root length density. The use of sophisticated electrical models, coupled to existing macroscopic root development and nutrient uptake models (e.g., Dunbabin et al., 2013; Javaux et al., 2013), could provide further insight to identify the key processes that control the electrical polarization signatures of roots.

As already pointed out by Repo et al. (2012), single-frequency measurements are of limited value to determine electrical polarization properties of root systems, both in terms of spatial distribution and polarization strength, and our spectral EIS

results (Figs. 5, 7, 12, and 13) support this finding. It becomes even more obvious when interpreting the polarization signature as EDL response, which typically exhibits a strong frequency dependence. It should be noted that the frequency range analyzed here in an imaging framework (0.46 Hz to 220 Hz) does not cover the full bandwidth that could be, in principle, measured with the presented setup, and corresponding advances are within easy reach (e.g., Huisman et al., 2015). However, reliable measurements at lower and higher frequencies will require careful adaptations in measurement and data processing procedures.

The classification of image pixels into two classes cannot, and should not, be treated as an universal analysis procedure. For the simple conditions in this experiment a clear distinction between root area and surrounding medium could be made, which facilitated the assessment of the method (e.g., Fig. 11), and can potentially be used for further experiments with root systems in aqueous solutions. However, the primary results of this study do not rely on this specific classification, and likewise soil-based

experiments could be conducted with the measurement setup.

This study does not involve any kind of granular substrate and thus excluded possible influences from such a background material. In fact, significant additional electrical polarization can be expected when soil surrounds the root system, which will superimpose on the root system response. Organic matter and micorrhiza may also contribute to the overall electrical signature. Finally, a variable water content can significantly influence the electrical response of the soil and the root system, either directly

by influencing present EDLs, or indirectly by inducing physiological processes such as nutrient uptake, which in turn can affect the electrical signatures of the EDLs.

## 5.2   Geophysical methodology

The observed polarization response of the root system is relatively weak and its measurement requires a corresponding accuracy of the measurement instrument. This accuracy is provided by the EIT-40 tomograph that was used in this study (Zimmermann

et al., 2008). The high accuracy of the instrument was recenty also demonstrated in an imaging study on soil columns (Kelter et al., 2015).

If a fixed data weighting is used, which we believe to produce more reliable and consistent results for multi-frequency time-lapse data, data selection, i.e. filtering, becomes a relevant step in the processing pipeline before the inversion and subsequent spectral analysis. While it is common to remove outliers from geophysical data prior to inversion, filtering becomes challenging

if multiple time steps are to be analyzed in a consistent way. The number of retained data points varied slightly between time steps, although the same filtering criteria were applied. This can be explained by data noise and varying contact impedances



at the electrodes. However, data quality was sufficient enough to produce consistent imaging results for all time steps and frequencies, as is evident from the impedance spectra (Fig. 5).

Another important issue is the data processing flow in the imaging framework, coupled with the spectral analysis based on the Debye decomposition. The inversion algorithm produces spatially smooth images; however, the images were computed for each frequency separately, and thus no smooth variation between adjacent frequencies is enforced in the inversion, although physically expected. Corresponding inversion algorithms have been developed recently (Kemna et al., 2014; Günther and Martin, 2016) and could lead to a further improvement of the multi-frequency imaging results. However, a similar constraint is introduced by the Debye decomposition, where smoothness is imposed along the relaxation time axis (which directly corresponds to the frequency axis). Minor noise components can thus be expected to be smoothed out both spatially and spectrally.

EIT applications in pseudo-2D rhizotron containers require specific processing steps. The determination and testing of correction factors accounting for modeling errors due to an imperfect 2D situation (Fig. 2) is as important as the correct representation of the rhizotron in terms of the FE grid underlying the inversion process (Fig. 3). Not taking these aspects into account can produce artefacts in the imaging results that can easily be misinterpreted in biological terms. Similarly, data errors should not be underestimated, as this can also produce misleading imaging results when data are overfitted (e.g., Kemna et al., 2012). Contrarily, an overestimation of data errors can mask information present in the data. Among others, raw (impedance) data and imaging (complex conductivity) data should be checked for consistency and plausibility by taking into account the much lower spatial resolution of the raw data (cf. Figs. 5 and 7).

Electrical imaging results exhibit a spatially variable resolution, which usually decreases with increasing distance from the electrodes. One could question the usefulness of such a method if the resolution cannot be clearly determined. Nonetheless, even limited spatial information allows for a distinction of polarizing and non-polarizing regions in the investigated object. This is not possible with spectroscopic measurements, and correspondingly it is more difficult with such measurements to analyze spatially distributed root systems. We suspect that some of the reported inconsistencies in electrical capacitance relationships with biological parameters (e.g., Kormanek et al., 2015, and references therein) can be ascribed to missing spatial information in the measurement data. The resolution of EIT is not sufficient to image microscopic current flow paths in the root system, but the imaged macroscopic electrical properties can be compared for different regions of the root system, for instance the older top part of the root system compared to the younger lower part. Future improvements in experimental setups (electrode distribution and spacing) and measurement configurations will most probably lead to increased spatial resolution.

Another advantage of the EIT approach presented in this study is the possibility of arbitrarily placed electrodes (as long as the resulting geometrical arrangement allows for a sufficient measurement coverage of the root system), in contrast to using stem electrodes as commonly done in previous studies. If the stem of a plant is used to inject current into the root system, measurements, and resulting correlations to biological parameters, are highly sensitive to the electrode position above the stem base (Dalton, 1995; Ozier-Lafontaine and Bajazet, 2005). Another problem is that electrodes cannot be inserted into the stem if damage of the plant is to be avoided. However, injections can also be realized by use of non-invasive clamps.

Also important for the experimental design is the time scale of the physiological response to be monitored. The spectral EIT measurements presented here took approximately three and a half hours to complete a single frame. Physiological pro-



cesses taking place on a shorter time span can thus not be resolved. Reducing the data acquisition time can be achieved by either reducing the number of low-frequency measurements or the number of current injections. This can result in a loss of spectral and spatial resolution if measurement configurations are not suitably optimized to compensate for the lost number of measurements.

# 6 Conclusions

The goal of this study was to investigate and establish spectral (i.e. multi-frequency) EIT as a non-invasive tool for the characterization and monitoring of crop root systems. Based on working hypotheses derived from the state of science in the involved fields, including geophysics and plant science, we designed and conducted a controlled experiment, in which the root systems of oilseed plants were monitored in a 2D, water-filled rhizotron container. Since water does not exhibit a significant polarization response in the considered frequency range, the observed electrical polarization response could be attributed to the root system.

The spectral EIT imaging results revealed a low-frequency polarization response of the root system, which enabled the successful delineation of the spatial extension of the root system. Based on a pixel-based Debye decomposition analysis of the spectral imaging results, we found a mean relaxation time of the root system's polarization signature in the covered frequency range of the order of 10 ms, corresponding to a frequency of the order of 15 Hz. Importantly, upon ongoing nutrient deprivation the magnitude of the overall polarization response steadily decreased and the spectral characteristics systematically changed, indicating changes in the length scales on which the polarization processes took place in the root system. The spectral EIT imaging results could be explained by the macroscopically observed and expected physiological response of the plant to the imposed nutrient deficiency. The identification of the root structures and processes controlling the root electrical signatures, however, was beyond the scope of this study given the inherent spatial resolution limits of EIT. Nonetheless the recovered electrical signatures could be used in the future to develop and calibrate improved macroscopic root electrical models which incorporate microscopic processes.

We showed, for the first time (to the best of our knowledge), that spectral EIT is a capable non-invasive method to image root system extension as well as to monitor changes associated with root physiological processes. Given the applicability of the method at both laboratory and field scale, our results suggest an enormous potential of spectral EIT for the structural and functional imaging of root systems for various applications. In particular at the field scale, non-invasive methods for root system characterization and imaging are lacking and EIT seems to be a very promising method to fill this gap. In future studies we will aim at further proving the suitability of spectral EIT to monitor physiological responses in different situations and to different stimuli, at both laboratory and field scales.



## 7  Acknowledgements

Parts of this work were funded by the Deutsche Forschungsgemeinschaft (DFG) in the framework of the project "Non-destructive characterization and monitoring of root structure and function at the rhizotron and field scale using spectral electrical

20   impedance tomography" (KE 1138/1-1) and the collaborative research centre "Patterns in soil-vegetation-atmosphere systems: monitoring, modeling and data assimilation" (SFB/TR 32). We are especially grateful to Egon Zimmermann and Matthias Kelter for valuable discussions regarding the measurement setup. We also thank Johannes Pfeifer and Achim Walter for technical support and discussions on root physiology in the initial phase of the work.





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
