# Peer review of "Multi-frequency electrical impedance tomography as a non-invasive tool to characterize and monitor crop root systems"

_Biogeosciences, 2016_

## Referee Comment (RC1) · Anonymous Referee #1 · 16 Nov 2016

I consider that the paper is now worth for publication because the authors present a fresh hypothesis that the electrical double layer appears at cell membrane, and this double layer contribute influence to the nutrient concentrations on both sides of membrane and give some contribution related to this topic.

---

## Author Comment (AC1) · 16 Nov 2016

Please note that the previously published PDF file of the discussion paper was corrupt (most of the text missing from the figures). This problem was solved with the help of the editorial team. Please re-download the PDF file in case your version contains broken figures.

---

## Referee Comment (RC2) · Anonymous Referee #3 · 24 Nov 2016

This is a very interesting paper using multi-frequency electrical impedance tomography as a non-invasive tool to monitor plant root activity.

I liked very much the introduction and also the second chapter, where a review on electrical polarization on roots and on polarization of biomatter is realized. I also appreciated the definition of working hypotheses.

The result section is in general clear and well written. I have some minor concerns though with the result plots, where labels, axes, captions are not always clear (this has been solved in the meantime:please do not consider all remarks on plots)

I had a problem to follow the discussion on the biological interpretation of the mea-

surements but maybe I misunderstood what authors meant. I think this section would benefit from a clearer explanation. Additional measurements, which would support these explanations should ideally also be provided or at least discussed. I only have minor comments, which are summarized here below.

Comments p.6, l.13: define hat zeta potential is

p.6, l.30: "the matter between" :in biology, this is called "root cortex"

p.7, l.20: this is the first time you mention cell wall. You should explain in the review before why it is important (it is not explicit in the text at the end of p.6).

p.7, l.20: define "plasma membrane": is it cell membrane?

p.8, l.33: is there a reason why the error model is only on the resistance on not on the phase as well?

p.9, l.24: replace "depth" by "thickness". Figure 1. In the plot caption, explain the meaning of the arrows and of the number or refer to the text.

p.11, l.16: is this correction independent on the background concentration?

Figure 3: put axis labels, delete the subplot titles and add a, b, and c

p.13, last line: give the lateral extent as well.

Figure 5. add units on the axes. Explain the numbers of the color line (i.e., hours and days?)

Figure 6. add subplot letters, units and axis labels. Add a scale on the photograph.

p.16, first lines of section 4.4.: could you show these area on one of the rhizotron photographs that you showed before?

p.16, l.5: do you mean thick and thin root segments? "Large" is not really accurate. Try also to avoid using the word "root" throughout the manuscript, which is sometimes used for a root system, sometimes for a root brench or sometimes for a root segment.

Clarify throughout the text.

Figure 7. same comments than before about subplot titles, letters and color legend. You should also add a gray zone here where results are below measurement accuracy.

Figure 11. Axis labels are not visible.

p.20, L.21: add "electrical" before conductivity

p20, L.21: add "solution" after "rhizotron"

p.20, l.22: root water uptake only cannot explain an increase of the salinity. This is only the case if solute uptake rate is lower than water uptake rate.

p.21, l. 13: "Accordingly…"I don't really follow the reasoning here. (1) A general increase of solute concentration is observed in the liquid phase (see end of previous page). (2) that would mean that there the water uptake is lower than the solute uptake (i.e. exclusion of solute): the extracted water by plant will thus have a lower concentration than the rhizotron water, which should lead to an increase of solute concentration around roots (due to exclusion). (3) an additional question is whether chemical diffusion in water does not counterbalance this gradient of concentration around roots. Yet, I am not sure what the authors mean by "outer root region": do you mean the root cortex? Or the part of the root zone which is at the edge of the root system? or do you mean the rhizosphere?

P21, L.16: "nutrient": is it on purpose that you speak about nutrient here and not about all solutes. Do you mean that some solutes are excluded by roots and that nutrients are taken proportionally or more than proportionally to water uptake? As a function of the ratio between nutrient and the other 'useless' solutes, that could result in an increase of the salinity but a decrease of the nutrient concentration (?) Please clarify

P21, l.20: it would be good to support this hypothesis of stress with data: nutrient concertation in the solution at the end of the experiment, salinity level, plant transpiration decrease with time, ... Do you have them? At least you should discuss that these

informations would have helped support your interpretation.

---

## Author Response (AR1)

2 MULTI-FREQUENCY ELECTRICAL IMPEDANCE TOMOGRAPHY AS A
3 NON-INVASIVE TOOL TO CHARACTERIZE AND MONITOR CROP ROOT
4 SYSTEMS

5 **Responses to review comments**

7 Dear Mr. Stoy, dear reviewers,

8 We thank the reviewers for their kind reviews and constructive comments. The com-
9 ments inspired incredibly helpful trains of thought, especially concerning the design of
10 future experiments.

11    In the revised version of the manuscript, we have considered all points raised by
12 the reviewers. In addition, we now discuss a potential anaerobic situation during the
13 experiment, along with the impact on measured electrical signatures. Also, we added
14 a link to the data package under a new section '7. **Data Availability**'.

15 Please find attached a version of the manuscript with all changes color-highlighted.
16 Our responses to the reviewer comments are listed on the next pages. Based on these
17 changes, we would be grateful if you would consider the manuscript for publication in
18 *Biogeosciences.*

19 Best regards,

20 Maximilian Weigand and Andreas Kemna

**Response to referee #3 (report #2)**

*1: I had a problem to follow the discussion on the biological interpretation of the measurements but maybe I misunderstood what authors meant. I think this section would benefit from a clearer explanation.*

Handled while responding to the comments down below.

*2: Additional measurements, which would support these explanations should ideally also be provided or at least discussed.*

Discussed in the last comment.

Indeed, some aspects of this section may have been hard to follow. We tried to address this issue in our changes to the section, detailed in the specific comments down below.

*3: p.6, l.13: define what zeta potential is*

Done. We changed the sentence:

*Recently, Li et al. (2015) determined Zeta-potentials of rice-root surfaces using elektrokinetic measurements, providing a relatively easy way to estimate the surface potentials of intact plant roots.*

to

Li et al. (2015) estimated the electric potential at rice-root surfaces of macroscopic root segments using measurements of the electrokinetic Zeta-potential. The Zeta-potential is the experimentally accessible electric potential at some distance from the surface where slipping in the electrolyte occurs upon a flow-driving pressure gradient.

*4: p.6, l.30: "the matter between" :in biology, this is called "root cortex"*

changed

*5: p.7, l.20: this is the first time you mention cell wall. You should explain in the review before why it is important (it is not explicit in the text at the end of p.6).*

Thank you very much for pointing out this important detail. In fact the cell walls should play only a minor role when electrical polarization properties of cells and cell clusters

are considered. We clarified this throughout the text and added the following sentence to the introduction:

*According to Kinraide et al. (1998) cell walls can be assumed to be near ionic equilibrium with the surrounding electrolyte and thus do not contribute to the formation of EDLs in biomaterial.*

6: *p.7, l.20: define "plasma membrane": is it cell membrane?*

Yes, indeed the terms were not clearly defined. We see both terms as synonyms and clarified this in the introduction

7: *p.8, l.33: is there a reason why the error model is only on the resistance and not on the phase as well?*

The tomographic inversion algorithm is formulated in the complex domain, i.e., both resistance (magnitude) and phase data are inverted simultaneously. Here, weighting of the data misfit is done by the (real-valued) magnitude of a complex-valued error estimate, which, however, is dominated by the resistance error since the phase values are relatively small for the measurements considered here. Therefore the resistance error model can be used in the complex inversion for the weighting of the complex data (including the phase). This is reflected in the changes made to section 3.1.2, where we now also provide a reference.

8: *p.9, l.24: replace "depth" by "thickness". Figure 1. In the plot caption, explain the meaning of the arrows and of the number or refer to the text.*

Done

9: *p.11, l.16: is this correction independent on the background concentration?*

Yes, for a homogeneous conductivity distribution the correction factors are theoretically independent of the conductivity value, i.e., the background concentration (since measured resistance and resistivity (inverse of conductivity) are linearly related for a homogeneous distribution). Significant changes in the correction factors can only occur for strong spatial conductivity variations, in particular across the thickness of the rhizotron (2D/3D effects). However, even if present, such effects in the correction factors would primarily result in inaccuracies in the inverted conductivity magnitude image, while the conductivity phase image, and also the DD-derived spectral parameters (total chargeability, relaxation time), are relatively robust against magnitude, i.e., correction factor, errors. We therefore, for the small to moderate conductivity variations observed in the experiment in the upper region of the rhizotron, assume that the conducted calibration survey, i.e., one universal set of correction factors, was actually sufficient. We added corresponding lines in section 3.4.2.

10: *Figure 3: put axis labels, delete the subplot titles and add a, b, and c*

Done

11: *p.13, last line: give the lateral extent as well.*

Done

12: *Figure 5. add units on the axes. Explain the numbers of the color line (i.e., hours and days?)*

Done (PDF was damaged)

13: *Figure 6. add subplot letters, units and axis labels. Add a scale on the photograph.*

Done

14: *p.16, first lines of section 4.4.: could you show these areas on one of the rhizotron photographs that you showed before?*

We now indicate the two locations near the root systems in Fig. 4a using colored dots. The third location is not indicated, as it lies below the covered region of the photograps.

15: *p.16, l.5: do you mean thick and thin root segments? "Large" is not really accurate. Try also to avoid using the word "root" throughout the manuscript, which is sometimes used for a root system, sometimes for a root brench or sometimes for a root segment. Clarify throughout the text.*

Done

16: *Figure 7. same comments than before about subplot titles, letters and color legend.*

Done

17: *You should also add a gray zone here where results are below measurement accuracy.*

Done

18: *Figure 11. Axis labels are not visible.*

Done (PDF was damaged)

19: *p.20, L.21: add "electrical" before conductivity*

Done.

20: *p20, L.21: add "solution" after "rhizotron"*

Done.

21: *p.20, l.22: root water uptake only cannot explain an increase of the salinity. This is only the case if solute uptake rate is lower than water uptake rate.*

Thank you very much for pointing this out. In fact, we revisited the spatial and temporal distribution of the conductivity changes in the whole rhizotron and noted that the origin of the conductivity increase is at the bottom of the rhizotron and thus not related to the root system. We fully agree that neither root water uptake nor evapotranspiration could cause this large increase in conductivity.

We added images of the conductivity distribution in the whole rhizotron to the supplement and discuss the issue in the first paragraph of section 5.1 (Biological Discussion).

22: *p.21, l. 13: "Accordingly. . ."I don't really follow the reasoning here. (1) A general increase of solute concentration is observed in the liquid phase (see end of previous page). (2) that would mean that there the water uptake is lower than the solute uptake (i.e. exclusion of solute): the extracted water by plant will thus have a lower concentration than the*

*rhizotron water, which should lead to an increase of solute concentration around roots*
*(due to exclusion). (3) an additional question is whether chemical diffusion in water does*
*not counterbalance this gradient of concentration around roots. Yet, I am not sure what*
*the authors mean by "outer root region": do you mean the root cortex? Or the part of the*
*root zone which is at the edge of the root system? or do you mean the rhizosphere?*

Thank you for pointing this out. We now restructured the corresponding section and
discuss possible interpretations considering the water potential gradient within the
root system, and the general lack of nutrient availability during the experiment. Thus,
we do not make any specific assumption regarding ion concentrations in the liquid
phase anymore, but provide possible cases in which either, by means of selective so-
lute uptake, ion concentrations decrease in the vicinity of charged surfaces within the
root system, or electrical surface characteristics change in reaction to the induced phys-
iological stress situation.

We believe that this reasoning is more consistent within the discussion, without
being too specific with respect to issues that can neither be proofed nor disproofed by
our study.

*23: P21, L.16: "nutrient": is it on purpose that you speak about nutrient here and not about*
*all solutes. Do you mean that some solutes are excluded by roots and that nutrients are*
*taken proportionally or more than proportionally to water uptake? As a function of the*
*ratio between nutrient and the other 'useless' solutes, that could result in an increase of*
*the salinity but a decrease of the nutrient concentration (?) Please clarify*

Our intention was to talk about solutes in general, with the intended interpretation
that the dynamics of nutrients in the system dominate the physiological response of
the plant system (in the absense of 'poisonous' ion families, all other solutes should
not play a significant role in the physiological processes of the plant).

*24: P21, l.20: it would be good to support this hypothesis of stress with data: nutrient*
*concentration in the solution at the end of the experiment, salinity level, plant transpi-*
*ration decrease with time, … Do you have them? At least you should discuss that these*
*informations would have helped support your interpretation.*

Unfortunately we do not have any additional data. While we fully acknowledge the
usefulness of these parameters, we also like to point out that this study is concerned
with the methodological establishment of using EIT for crop root research. We be-
lieve that at this stage the potential usefulness of the method could be shown without
additional environment data.

We added one sentence to the discussion mentioning the usefulness of those pa-

159  rameters in future experiments.

[revised manuscript text omitted]